# Manipulating Li$_2$S$_2$/Li$_2$S mixed discharge products of all-solid-state lithium sulfur batteries for improved cycle life

Jung Tae Kim ®[1], Adwitiya Rao ®[2], Heng-Yong Nie ®[3,4], Yang Hu[1], Weihan Li[1], Feipeng Zhao ®[1], Sixu Deng[1], Xiaoge Hao[1], Jiamin Fu[1], Jing Luo[1], Hui Duan[1], Changhong Wang ®[1,5] ✉, Chandra Veer Singh ®[2] ✉ & Xueliang Sun ®[1,5] ✉

All-solid-state lithium-sulfur batteries offer a compelling opportunity for next-generation energy storage, due to their high theoretical energy density, low cost, and improved safety. However, their widespread adoption is hindered by an inadequate understanding of their discharge products. Using X-ray absorption spectroscopy and time-of-flight secondary ion mass spectrometry, we reveal that the discharge product of all-solid-state lithium-sulfur batteries is not solely composed of Li$_2$S, but rather consists of a mixture of Li$_2$S and Li$_2$S$_2$. Employing this insight, we propose an integrated strategy that: (1) manipulates the lower cutoff potential to promote a Li$_2$S$_2$-dominant discharge product and (2) incorporates a trace amount of solid-state catalyst (LiI) into the S composite electrode. This approach leads to all-solid-state cells with a Li-In alloy negative electrode that deliver a reversible capacity of 979.6 mAh g$^{-1}$ for 1500 cycles at 2.0 A g$^{-1}$ at 25 °C. Our findings provide crucial insights into the discharge products of all-solid-state lithium-sulfur batteries and may offer a feasible approach to enhance their overall performance.

The increasing number of countries committing to net-zero emissions has sparked a greater demand for economically feasible, highly energy-dense, and intrinsically safe energy storage systems[1,2]. All-solid-state lithium-sulfur batteries (ASSLSBs) have emerged as a promising energy storage solution because they possess several distinct advantages compared to traditional electrochemical energy storage systems such as lithium-ion batteries (LIBs). First, ASSLSBs utilize abundant, evenly distributed, and cost-effective sulfur as the active material[3,4]. Second, the high specific capacity of sulfur (1672 mAh/g) and lithium metal (3860 mAh/g) offers a theoretical specific energy of 2600 Wh/kg, which is much higher than traditional LIBs[3,4]. Third, ASSLSBs replace the flammable liquid electrolyte with a non-flammable inorganic solid-state electrolyte (SSE), mitigating the thermal runaway concerns

inherent to traditional liquid electrolyte-based batteries[5,6]. All-solid-state configurations also eliminate the polysulfide shuttle effect, a phenomenon that is notorious for plaguing the development of liquid lithium-sulfur batteries (LSBs)[7–9].

Due to their numerous advantages, tremendous efforts have been dedicated to the development of ASSLSBs, particularly using sulfide-based solid-state electrolytes (SSEs) because of their high room-temperature ionic conductivity and low mechanical moduli[10–12]. However, despite ongoing efforts, ASSLSB technology remains nascent, and several challenges prevent it from surpassing the specific energy capabilities of current LIBs and LSBs. Some of these challenges include: (1) low electronic/ionic conductivity of S/Li$_2$S resulting in poor active material utilization; (2) severe volume changes of sulfur (~78%)

[1]Department of Mechanical and Materials Engineering, University of Western Ontario, 1151 Richmond St, London, Ontario, ON N6A 3K7, Canada. [2]Department of Materials Science and Engineering, University of Toronto, Ontario, ON M5S 3E4, Canada. [3]Surface Science Western, University of Western Ontario, 999 Collip Circle, London, Ontario, ON N6G 0J3, Canada. [4]Department of Physics and Astronomy, University of Western Ontario, 1151 Richmond St, London, Ontario, ON N6A 3K7, Canada. [5]Eastern Institute for Advanced Study, Eastern Institute of Technology, Ningbo, Zhejiang 315200, P.R. China. ✉e-mail: changhongwang@eias.ac.cn; chandraveer.singh@utoronto.ca; xsun9@uwo.ca

upon (de)lithiation causing physical contact losses and poor reversible redox; (3) SSE degradation leading to the formation of less conductive interphases that increase interfacial resistance and hinder electron/ion transport in the S composite electrode; and (4) lithium dendrite growth that causes short-circuiting and substantially diminishes battery lifetime as a result[3,7,8,13]. Various strategies have been adopted to address the challenges mentioned above and improve the specific energy of ASSLSBs, such as nanosizing S/$Li_2S$ to establish more triple-phase boundaries and improve active material utilization[14,15], forming solid solutions to improve redox reversibility[16,17], and suppressing lithium dendrite growth by interface modification[18–20].

While these strategies have proven fruitful, a key obstacle hindering the development of ASSLSBs stems from conceptual ambiguity surrounding their underlying redox mechanisms. Initial research employing in situ transmission electron microscopy explored the evolution of $Li_2S$ in ASSLSBs, revealing a three-step lithiation process and direct conversion from $S_8$ to $Li_2S$, without the formation of other sulfur species[21]. Another study investigated the decomposition behavior of $Li_2S$ highlighting that the decomposition of $Li_2S$ is governed by $Li^+$ ion conductivity rather than electronic conductivity[22]. A recent study investigating the electrochemical reaction pathway of ASSLSBs reported the presence of a $Li_2S_2$ intermediate phase during the conversion from $S_8$ to $Li_2S$[23]. These studies have set important precedents and resulted in a richer understanding of the fundamental redox mechanisms of ASSLSBs. However, the intricate interplay between the discharge products and the electrochemical behavior of ASSLSBs, encompassing crucial aspects such as initial discharge capacity, cycling stability, and reversibility, remains insufficiently explored but stands as a pivotal prerequisite for driving the advancement of ASSLSB technology.

In this study, we first interpret the active material utilization of ASSLSBs reported in recent literature[20,24–32] to postulate a mixed discharge product consisting of lithium sulfide ($Li_2S$) and lithium disulfide ($Li_2S_2$). Using X-ray absorption spectroscopy (XAS), we reveal the existence of $Li_2S_2$, and confirm the premise of a mixed discharge product as a result. To achieve direct chemical identity of $Li_2S_2$ and provide further evidence of its existence, we use time-of-flight secondary ion mass spectrometry (ToF-SIMS)[33], which has superior chemical selectivity via the detection of diagnostic ions[34,35] and/or exploration of the relationships[36] among relevant ions. ToF-SIMS has previously been used to depth profile S⁻ in cycled sulfur cathodes of liquid Li-S batteries[37]. We demonstrate that ToF-SIMS can be used to detect $Li_2S_2$ and differentiate it from $Li_2S$, which can provide information of the relationship between the discharge products of ASSLSBs. Furthermore, using first principles calculations, we demonstrate that $Li_2S_2$ exhibits better redox kinetics than $Li_2S$, which suggests that the reversibility and cycling stability of ASSLSBs can be improved by inducing a $Li_2S_2$-dominant discharge product. As a proof-of-concept, we manipulate the lower cutoff voltage to induce a $Li_2S_2$-dominant final discharge product and ensure stable, long-term cycling performance. To further improve the conversion efficiency of ASSLSBs, a trace amount of lithium iodide (LiI) is incorporated into the S composite electrode to facilitate the electrochemical oxidation of $Li_2S_2$/$Li_2S$ during charge. As a result, ASSLSBs are fully reversible, and deliver a reversible capacity of 979.6 mAh g⁻¹ for 1500 cycles under a high specific current of 2.0 A g⁻¹, representing unrivaled cycling behavior for elemental sulfur positive electrodes in an all-solid-state configuration. To demonstrate their practical viability, LiI-incorporated ASSLSBs are tested at −10 and 60 °C, delivering stable cycling stability. High active material loading ASSLSBs also deliver areal capacities above 4.0 mAh cm⁻². This work provides valuable insights into the discharge product of ASSLSBs, and demonstrates a feasible approach toward achieving fully reversible, all-climate ASSLSBs with high capacity, long lifetime, and improved safety.

## Results and discussion
### Postulating a $Li_2S_2$/$Li_2S$ mixed discharge product
Figure 1a depicts the theoretical discharge curve of an ASSLSB showing a discharge capacity of 1672 mAh g⁻¹ that corresponds to an active material utilization of 100%. The initial discharge capacities of ASSLSBs reported in recent literature often falls below 1400 mAh g⁻¹, as summarized in Fig. 1b[20,24–32]. It should be noted that these values include capacity contribution that comes from sulfide SSE decomposition[38–40]. The single plateau observed in the charge/discharge curves of ASSLSBs has been widely attributed to a single-phase solid-solid conversion from $S_8$ to $Li_2S$, where the discharge product consists solely of $Li_2S$. Theoretically, if this premise holds true, the reported discharge capacities of ASSLSBs in literature should approach or even surpass sulfur's theoretical value (i.e.,1672 mAh g⁻¹), particularly when considering sulfide SSE decomposition.

Low discharge capacities observed in ASSLSBs can be interpreted three ways. First, sulfur's poor electronic/ionic conductivity results in a large quantity of unreacted sulfur that remains after discharge. While unreacted sulfur leads to low discharge capacity, it seems unlikely that only a very small amount of sulfur can participate in redox considering the large fraction of SSE and conductive additives that are typically used to fabricate sulfur composite electrodes (Supplementary Table 1). Second, solid-solid conversion from S to $Li_2S$ is restricted by a lower potential limit that is set too high. To verify this reason, we investigated the electrochemical behavior of ASSLSBs at different lower potential limits (Fig. 1c). The theoretical capacity of sulfur cannot be reached even when the discharge potential goes down to −0.2 V (Li-In/Li⁺). This result demonstrates that the lower potential is not the main reason for low discharge capacities observed in ASSLSBs. Third, the final discharge product of ASSLSBs is a mixture of $Li_2S_2$ and $Li_2S$. It is well known that the conversion of $S_8$ to $Li_2S_2$ contributes 50% theoretical capacity (836 mAh g⁻¹) and subsequent conversion of $Li_2S_2$ to $Li_2S$ contributes another 50% capacity (836 mAh g⁻¹). The premise of a $Li_2S_2$/$Li_2S$ mixed discharge product is reasonable considering the plethora of ASSLSBs studies that report discharge capacities between 836 mAh g⁻¹ and 1672 mAh g⁻¹.

Another recurrent feature observed in the literature regarding ASSLSBs is their poor electrochemical reversibility, particularly following the initial discharge cycle[15,20,41]. This phenomenon has been attributed to the irreversible formation of $Li_2S$, where the stable anti-fluorite structure of $Li_2S$ necessitates high activation potentials, typically approaching 4 V (versus Li⁺/Li), to facilitate the electrochemical oxidation (or delithiation) of $Li_2S$ back to $S_8$ during the charging process[42–45]. We conducted density functional theory (DFT) calculations to investigate the influence of $Li_2S_2$ and $Li_2S$ on the reversibility of ASSLSBs (Supplementary Note 1). The calculated formation energies of $Li_2S_2$ and $Li_2S$ were approximately −1.01 eV/atom and −1.59 eV/atom, respectively (Fig. 1d). These results indicate that $Li_2S_2$ exhibits better redox activities compared to $Li_2S$. However, it is important to note that the formation energy of $Li_2S_2$ remains considerably lower than that of $S_8$. This suggests that both $Li_2S_2$ and $Li_2S$ hinder the electrochemical reversibility of ASSLSBs. Previous studies have demonstrated the use of lithium iodide (LiI) to effectively enhance the electrochemical oxidation of $Li_2S$, thereby achieving fully reversible ASSLSBs[16,46,47]. Indeed, our DFT calculations reveal that the molecular conversion of $Li_2S_2$/$Li_2S$ to $S_8$ on the LiI(100) surface requires a lower activation barrier compared to the process in vacuum (Fig. 1d, e). These results suggest that LiI can facilitate the electrochemical oxidation of not only $Li_2S$ but also $Li_2S_2$, thereby improving the reversibility of ASSLSBs as a result. Further discussion regarding the DFT calculations and the role of LiI in promoting the electrochemical oxidation of $Li_2S_2$/$Li_2S$ is provided in Supplementary Note 1.

In the following section, we thoroughly investigate the discharge products of ASSLSBs and evaluate the impact of incorporating LiI to

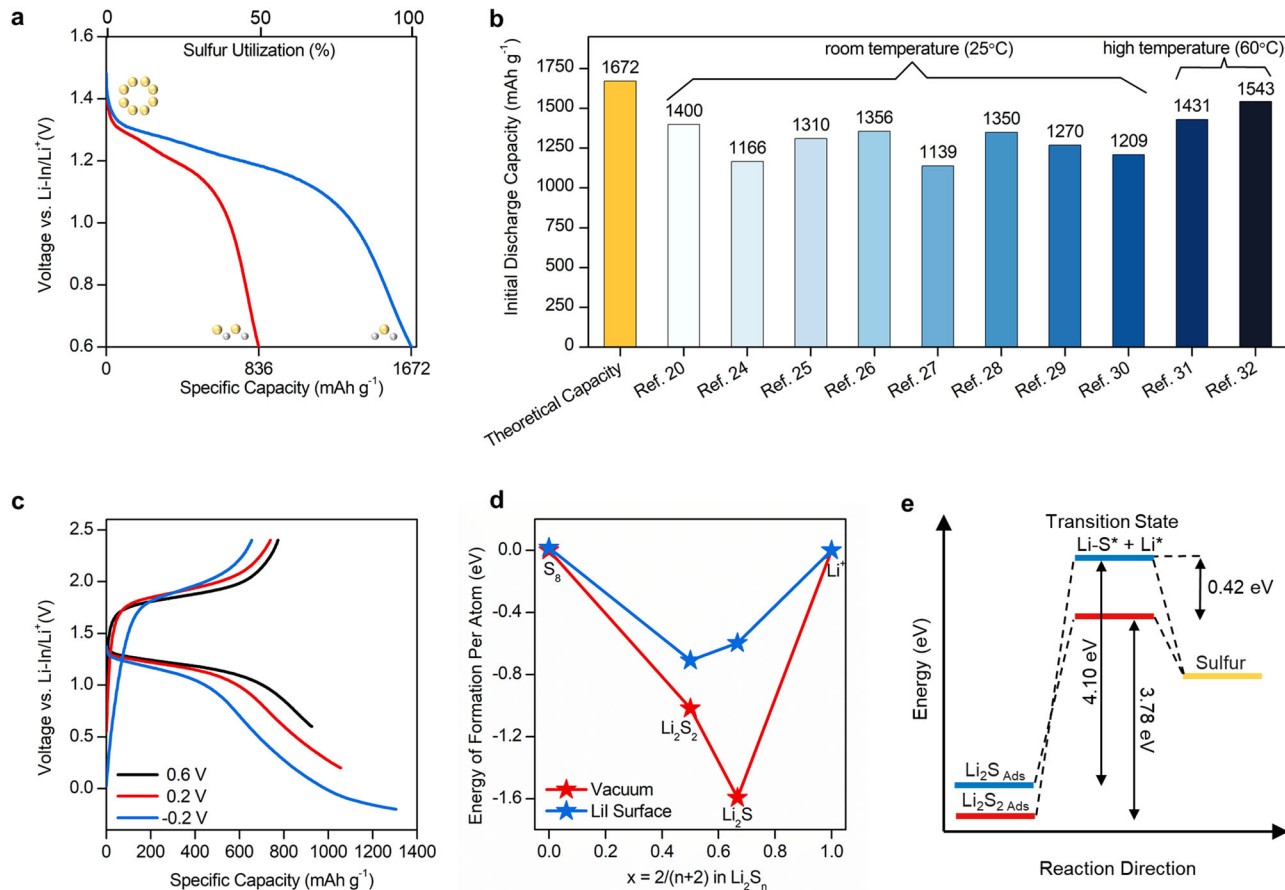

**Fig. 1 | Postulating a mixed discharge product of ASSLSBs. a** Voltage profile showing the theoretical discharge curve of ASSLSBs. **b** Initial discharge capacities of ASSLSBs recently reported in literature. **c** Voltage profile of an ASSLSB tested using different lower limit potentials. **d** Computational Gibbs-free formation energies of $Li_2S_2$ and $Li_2S$ per atom in vacuum (red line) and on the LiI(100) surface (blue line). **e** Simplified activation diagram illustrating the energy landscape of $Li_2S_2$ (red line) and $Li_2S$ (blue line) oxidation in the adsorbed phase on the LiI surface.

address the poor reversibility of ASSLSBs. For this study, two types of S composite electrodes were prepared and tested: one without LiI (S/ LGPS/CNT) and one with LiI (S/LGPS/CNT/LiI). Structural characterization and analysis of the S composite electrodes with and without LiI are presented in Supplementary Fig. 1–7 and Supplementary Note 2.

**Probing the discharge products of all-solid-state Li-S batteries**

Confirming the premise of a $Li_2S_2/Li_2S$ mixed discharge product is essential for providing insights into the reaction mechanism of ASSLSBs. Because $Li_2S_2$ exists as a meta-stable phase, determining its existence requires characterization techniques that are element-specific and chemically sensitive[48]. Synchrotron X-ray absorption spectroscopy (XAS) has been effectively used to identify and study various lithium polysulfide intermediates (i.e., $Li_2S_2$) in liquid/semi-liquid Li-S batteries[45,49,50]. In this study, XAS is used to determine the final discharge product of ASSLSBs by probing sulfur evolution at different discharge/charge states.

Figure 2a shows the S *K*-edge X-ray absorption near-edge structure (XANES) spectra of S composite electrode without LiI (i.e., S/LGPS/ CNT) at the pristine, fully discharged (100% DOD), and fully charged (100% SOC) state. The S *K*-edge XANES spectra of the S/LGPS/CNT composite before discharge (i.e., pristine state) shows two broad features at 2473.0 eV and 2480.0 eV, which correspond to elemental sulfur[51–53]. After discharge, three features at 2474.1, 2476.8 and 2484.4 eV emerge, which denote the partial formation of $Li_2S$[51,54,55]. Interestingly, a pre-edge feature appears at 2471.3 eV, which has previously been characterized as $Li_2S_2$ (Fig. 2b)[50,51,56]. After charge, the $Li_2S$

and $Li_2S_2$ features become weaker but are still present in the spectra, which indicates the irreversible transformation from $Li_2S_2/Li_2S$ to S. The XANES spectra of the LiI-incorporated composite at the pristine, fully discharged (100% DOD), and fully charged (100% SOC) state is plotted in Fig. 2c, d. After full charge, the $Li_2S$ and $Li_2S_2$ features become less prominent, and the features around 2473.0 eV and 2480.0 eV dominate again, resembling the pristine state (i.e., before discharge). This result suggests that LiI, even in trace quantities, plays a critical role in facilitating the electrochemical oxidation of $Li_2S_2/Li_2S$ during charge. The reversibility difference between the ASSLSBs at different discharge/charge states with and without LiI is further illustrated by the XANES spectra shown in Fig. 2e, f and Supplementary Fig. 8.

As for chemical analyses of $Li_2S_2$ and $Li_2S$, X-ray photoelectron spectroscopy (XPS) has been used previously to investigate the chemical composition of $Li_2S_2$ and $Li_2S$ in liquid Li-S batteries[57,58]. The detection of a S $2p_{3/2}$ peak at 162.2 eV is attributed to $Li_2S_2$ due to its binding energy's proximity to the reference sample of $Na_2S_2$ (162.0 eV). We utilized XPS to complement the XANES results and confirm the presence of the $Li_2S_2$ phase. However, as illustrated in Supplementary Fig. 9, a discernible $Li_2S_2$ peak was not found. Previous studies that investigate ASSLSBs using XPS show similar results, where no distinct $Li_2S_2$ peak is evident in the XPS spectra[32,59]. Chemical similarity and overlapping peaks of $Li_2S_2$ and $Li_2S$ pose challenges in accurately identifying $Li_2S_2$ using traditional XPS analysis. Additionally, the difficulty in isolating $Li_2S_2$ as a reference sample further complicates the analysis.

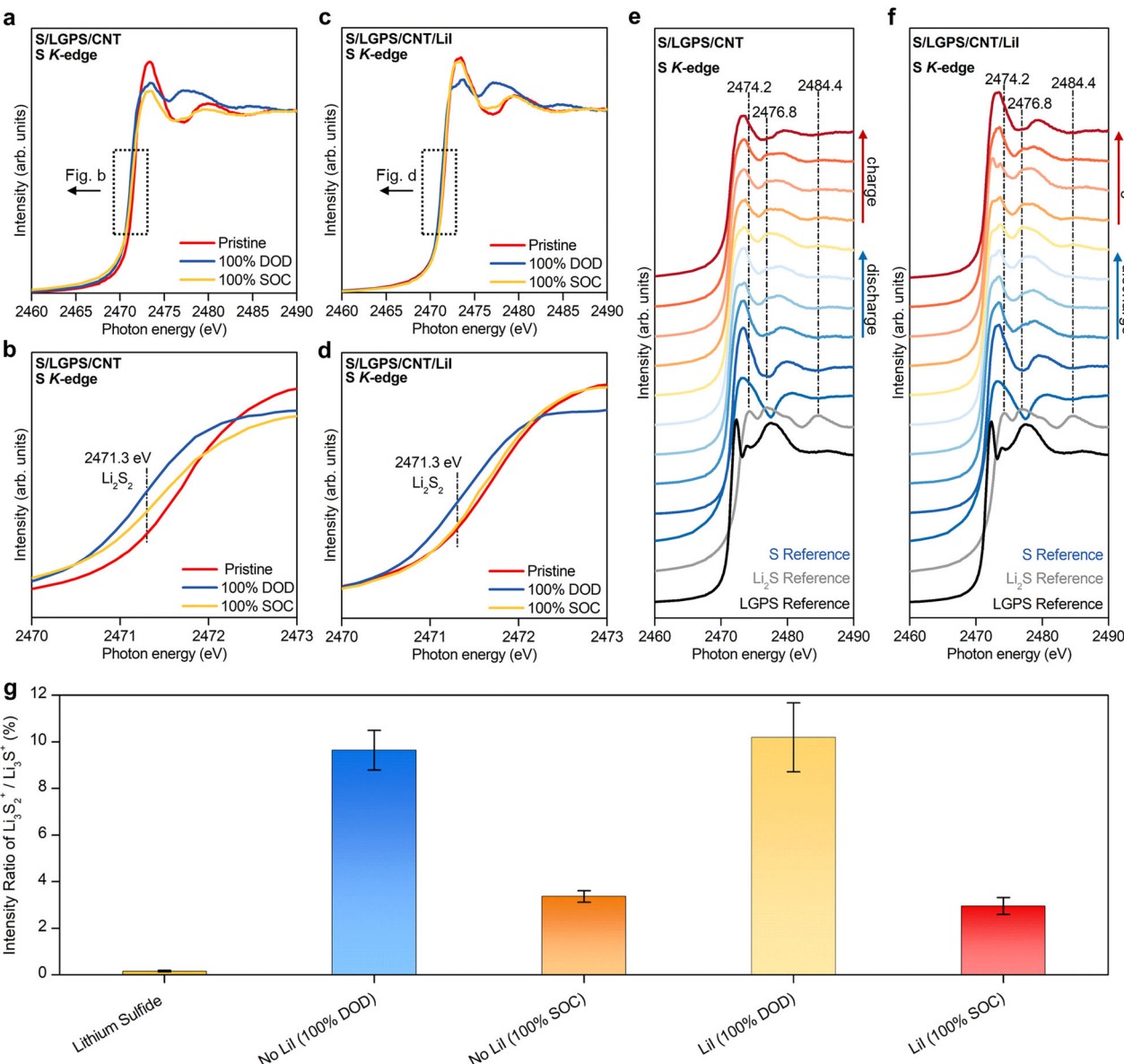

**Fig. 2 | Probing the final discharge products of ASSLSBs via X-ray absorption spectroscopy and time-of-flight secondary ion mass spectrometry. a, b** S K-edge XANES spectra of S composite electrodes without LiI. **c, d** S K-edge XANES spectra of LiI-incorporated S composite electrodes. **e** S K-edge XANES spectra of S scomposite electrode without LiI at different discharge/charge states. **f** S K-edge XANES spectra of LiI-incorporated S compossite electrode at different discharge/charge states. **g** Ion intensity ratio of $Li_3S_2^+/Li_3S^+$ for pure $Li_2S$ powder and the 100% DOD and 100% SOC products of ASSLSBs. The error bars represent the standard deviation of the measured intensity ratio and are produced using five independent measurements. DOD and SOC represent depth of discharge and state of charge, respectively.

Therefore, to directly determine the chemical identity of $Li_2S_2$ and gather additional supporting evidence for its existence, we used time-of-flight secondary ion mass spectroscopy (ToF-SIMS), which demonstrates superior chemical selectivity compared to XPS, enabling effective differentiation between $Li_2S$ and $Li_2S_2$. Both negative and positive secondary ion mass spectra were collected from pure $Li_2S$ powder (as reference) and the S composite electrodes with and without the addition of LiI. Ions related to the S composite electrodes include $Li^\pm$, $S_2^-$, $S_3^-$, $LiS^-$, $LiS_2^-$, $Li_2S^+$, $Li_3S^+$ and $Li_3S_2^+$. $Li_3S^+$ and $Li_3S_2^+$ are the most useful ions in differentiating $Li_2S$ and $Li_2S_2$ because they correspond to positive ions with the addition of a $Li^+$ to the molecules, i.e., $[Li_2S + Li]^+$ and $[Li_2S_2 + Li]^+$, respectively. While $Li_3S^+$ should be generated from both $Li_2S$ and $Li_2S_2$, $Li_3S_2^+$ is more likely generated from $Li_2S_2$. This ion fragmentation pattern was confirmed by looking at the positive ions of the pure $Li_2S$ powder, where there was little to no

$Li_3S_2^+$ detected. The positive secondary ion mass spectra in the mass range showing $Li_3S^+$ and $Li_3S_2^+$ are shown in Supplementary Fig. 10. Also shown in Supplementary Fig. 10 are the detection of $Li_2I^+$ and $LiI_2^-$ from the S composite electrodes with LiI added.

Shown in Fig. 2g are the ratios of the intensity of $Li_3S_2^+$ against $Li_3S^+$, which can be used to compare the relative portion of $Li_2S_2$ in the mixture of $Li_2S$ and $Li_2S_2$. For the pure $Li_2S$ powder, the ratio between $Li_3S_2^+$ and $Li_3S^+$ is practically zero as little to no $Li_3S_2^+$ is detected. For S composite electrode samples LiI 100% DOD, LiI 100% SOC, No LiI 100% DOD and No LiI 100% SOC, their ratios of $Li_3S_2^+/Li_3S^+$ are $10.2\% \pm 1.5\%$, $3.0\% \pm 0.4\%$, $9.6\% \pm 0.9\%$ and $3.4\% \pm 0.2\%$, respectively. Therefore, the ToF-SIMS results confirm that there is significantly more $Li_2S_2$ in the fully discharged (100% DOD) products of ASSLSBs than in the fully charged (100% SOC) ones, with or without the addition of LiI. With the addition of LiI, the fully charged

product of ASSLSBs shows a slightly reduced ratio in comparison with that without the addition of LiI.

## Inducing a $Li_2S_2$-dominant discharge product to enhance performance

After confirming the existence of a $Li_2S_2/Li_2S$ mixed discharge product, we devise an integrated strategy to enable high performance ASSLSBs by: (1) manipulating the lower potential to induce a $Li_2S_2$-dominant final discharge product and (2) incorporating a trace amount of LiI to facilitate the electrochemical oxidation of $Li_2S_2/Li_2S$. In principle, this approach should significantly enhance the reversibility and cycling stability of ASSLSBs. As a proof of concept, ASSLSBs with and without LiI were evaluated using a Li-In negative electrode and LGPS as the SSE interlayer (Fig. 3a). The lower voltage limit was set at 0.6 V (vs. Li-In/Li$^+$) to limit $Li_2S$ formation and obtain a $Li_2S_2$-dominant discharge product. ToF-SIMS analysis reveals that the intensity of $Li_3S^+$ ions decreased considerably in the cell discharged to 0.6 V compared to the one discharged to −0.2 V, which suggests that a $Li_2S_2$-dominant discharge product can be obtained by limiting the lower voltage threshold (Supplementary Fig. 11). Although promoting a discharge product with an $Li_2S_2$-dominant phase comes at the expense of the initial discharge capacity, doing so enhances both the reversibility and cycling stability of ASSLSBs (Supplementary Fig. 12). This is because $Li_2S_2$ is more electrochemically active than $Li_2S$, and the volumetric expansion of $Li_2S_2$ is comparatively smaller, at ~60%, compared to $Li_2S$ which expands by approximately 78%. As for fixing the optimum quantity of LiI to facilitate the electrochemical oxidation of $Li_2S_2/Li_2S$, we determined that a minimum of 6 wt% LiI was necessary to attain fully reversible ASSLSBs when setting the lower voltage limit to 0.6 V (vs. Li-In/Li$^+$). Lowering the LiI content to 3 wt% resulted in ASSLSBs that

could only be charged to approximately 86% of discharge capacity (Supplementary Fig. 13).

The ASSLSB without LiI loses approximately 18% of its initial discharge capacity after charging while the LiI-incorporated ASSLSB is fully reversible and exhibits a smaller electrode polarization (Fig. 3b). These results suggest that LiI plays a critical role in facilitating the electrochemical oxidation of $Li_2S_2$ and the small fraction of irreversibly formed $Li_2S$ after initial discharge, which coincide well with the DFT calculations presented in Fig. 1e, f. A galvanostatic intermittent titration technique (GITT) was used to estimate the Li$^+$ diffusion coefficient ($D_{Li}$) and evaluate the reaction kinetics of the ASSLSBs with and without LiI. The average $D_{Li}$ value for the LiI-incorporated ASSLSB during both the discharge and charge process is $4.83 \times 10^{-13}$ cm$^2$ s$^{-1}$, which is much higher than its counterpart (Supplementary Fig. 14). The LiI-incorporated ASSLSBs exhibits a reversible capacity of 100% during the GITT test, while the ASSLSB without LiI can only charge back to approximately 75%, further demonstrating the effectiveness of LiI for improving the reversibility of ASSLSBs. Rate performance of the ASSLSBs with and without LiI was investigated over a specific current range of 0.2 to 6.0 A g$^{-1}$ as shown in Fig. 3c The LiI-incorporated ASSLSBs delivers a discharge capacity of 933, 1027.4, 996.4, 978.9, 938.2, 760.8, 467.8, and 303.7 mAh g$^{-1}$ at specific current 0.2, 0.4, 0.6, 0.8, 1.0, 2.0, 4.0, and 6.0 A g$^{-1}$, respectively, recovering to 1222.4 mAh g$^{-1}$ as the specific current is restored back to 0.2 A g$^{-1}$. The ASSLSB without LiI delivers much lower discharge capacities in the subsequent cycles, due to the poor reversibility of the cell after initial discharge. The LiI-incorporated ASSLSBs also demonstrate much better stability to specific current changes and exhibit a smaller electrode polarization compared to the ASSLSB without LiI (Supplementary Fig. 15). These results suggest that LiI plays

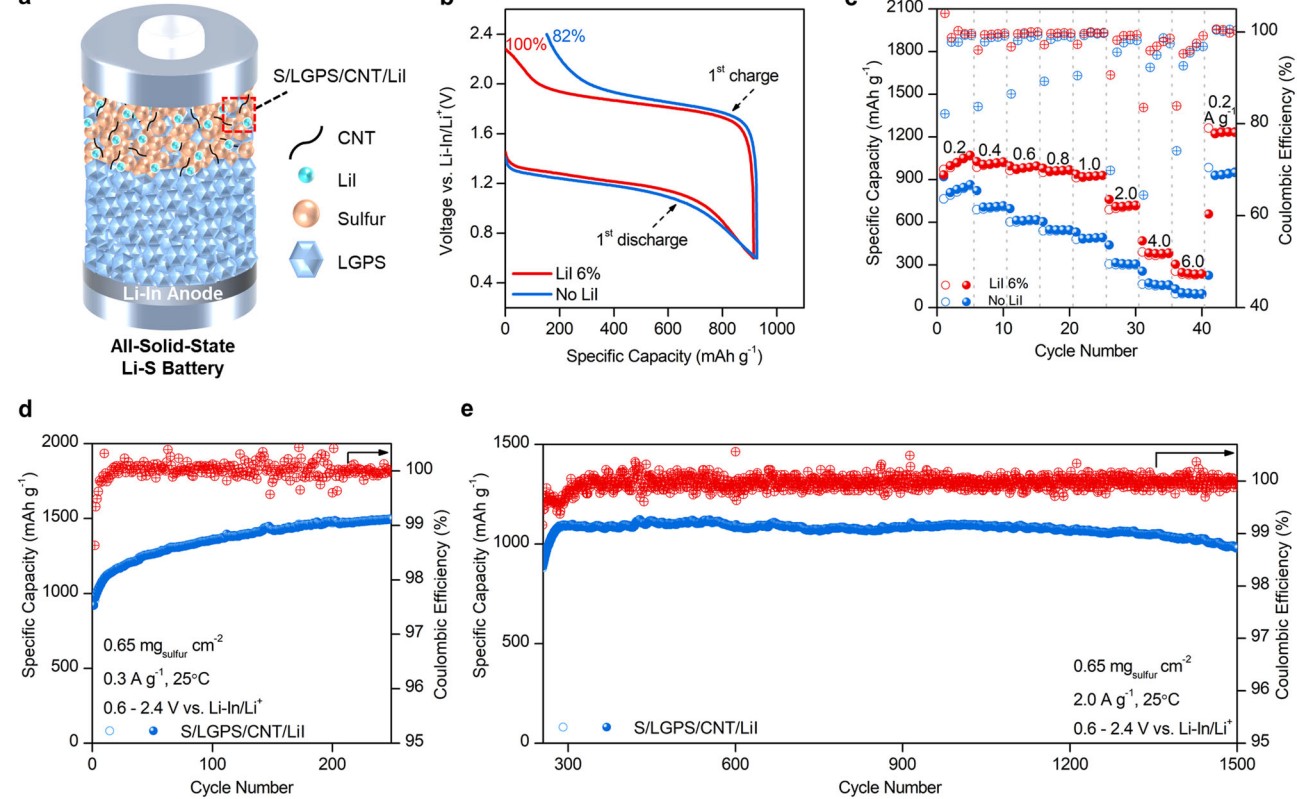

**Fig. 3 | Electrochemical behavior of all-solid-state Li-S batteries with $Li_2S_2$-dominant discharge product. a** Schematic illustration showing all-solid-state lithium-sulfur battery configuration. **b** Voltage profile showing the reversibility of ASSLSBs with and without LiI in the first cycle at 0.2 A g$^{-1}$ and 25 °C. **c** Rate performance comparison of ASSLSBs with and without LiI at different specific currents from 0.2 to 6.0 A g$^{-1}$ and 25 °C. **d** Cycling performance of ASSLSBs at 0.3 A g$^{-1}$ and 25 °C. **e** Continued cycling of ASSLSBs at 2.0 A g$^{-1}$ and 25 °C.

a vital role in enhancing charge transfer kinetics within the S composite electrode.

Figure 3d shows the cycling performance of the LiI-incorporated ASSLSB cycled between 0.6–2.4 V (vs. Li-In/Li+) at 0.3 A g−1 and 25 °C. A reversible capacity of 1496.9 mAh g−1 is obtained after 250 cycles. The gradual capacity increase observed during cycling is likely a result of two factors. First, sulfur undergoes an activation process in the initial cycles, as a large electrode polarization results in low active material utilization. As cycling continues, the electrode polarization decreases, resulting in higher active material utilization and increasing capacity. Indeed, the electrode polarization of the LiI-incorporated ASSLSB decreases from 0.721 V in the 1st cycle to 0.682 V in the 200th cycle (Supplementary Fig. 16). Similar behavior has been reported in other sulfur-based cathodes[60–62]. Second, thiophosphate SSEs such as LGPS possess a narrow electrochemical stability window (e.g., 1.71–2.14 V vs. Li+/Li), and decompose in the operating voltage range of ASSLSBs as a result[63]. The decomposition products of LGPS are electrochemically active, and contribute to the reversibility capacity of the cell[13]. These two phenomena can help explain the gradual capacity increase observed in Fig. 3d. To demonstrate long-term cycling, LiI-incorporated ASSLSBs were further cycled at a high specific current of 2.0 A g−1, delivering a stable capacity of 1069.4 mAh g−1 for over 1200 cycles and a reversible capacity of 979.6 mAh g−1 for over 1500 cycles (Fig. 3e). To our knowledge, the reported cycling behavior is the best to date for elemental sulfur cathodes in an all-solid-state configuration (Supplementary Fig. 17).

## All-climate all-solid-state Li-S batteries

Developing ASSLSBs that can operate within a wide temperature range is crucial for enabling applications such as electric aviation, electric vehicles, and spaceflight[64]. Thus, LiI-incorporated ASSLSBs were

further evaluated at high and low temperature to evaluate their practical viability. When tested at 60 °C, the cell shows much lower overpotential, and delivers a high initial discharge capacity of 1136.8 mAh g−1 (Fig. 4a). Interestingly, another discharge plateau appears at approximately 1.4 V (vs. Li-In/Li+). At 25 °C, the stepwise transition from elemental sulfur to high order polysulfides, low order polysulfides, and finally Li2S is not obvious in a solid-state configuration, as a high conversion barrier results in sluggish conversion kinetics. However, charge transfer within the S composite electrode is improved under more favorable conditions such as at elevated temperature, allowing for stepwise sulfur redox to occur. This likely gives rise to a distinct discharge plateau in the voltage profile at 60 °C, corresponding to the formation of intermediate sulfur species. A similar phenomenon has been observed in ASSLSBs that incorporate selenium into the sulfur cathode and solid-state lithium-selenium batteries, which is logical considering the high conductivity of selenium[62,65].

The cycling stability of the LiI-incorporated ASSLSB at 60 °C and 0.4 A g−1 is shown in Fig. 4b. The cell delivers a reversible capacity of 1323.6 mAh g−1 for over 400 cycles, demonstrating stable cycling stability at 60 °C. The capacity of the high temperature cell is much higher than the one tested at 25 °C, which corresponds to a greater quantity of Li2S that is formed after discharge. At high temperature, charge transfer kinetics within the S composite electrode improves, and sulfur redox can occur more efficiently as a result. Consequently, the solid-solid conversion of Li2S2 to Li2S is less hindered, leading to higher initial discharge capacities.

ASSLSBs with active material loadings for of 3 and 12 mg cm−2 were tested to evaluate the practical viability of the cells, as shown in Fig. 4c and Supplementary Fig. 18, respectively. Both cells are fully reversible and sustain areal capacities around 3.0 mAh cm−2 for

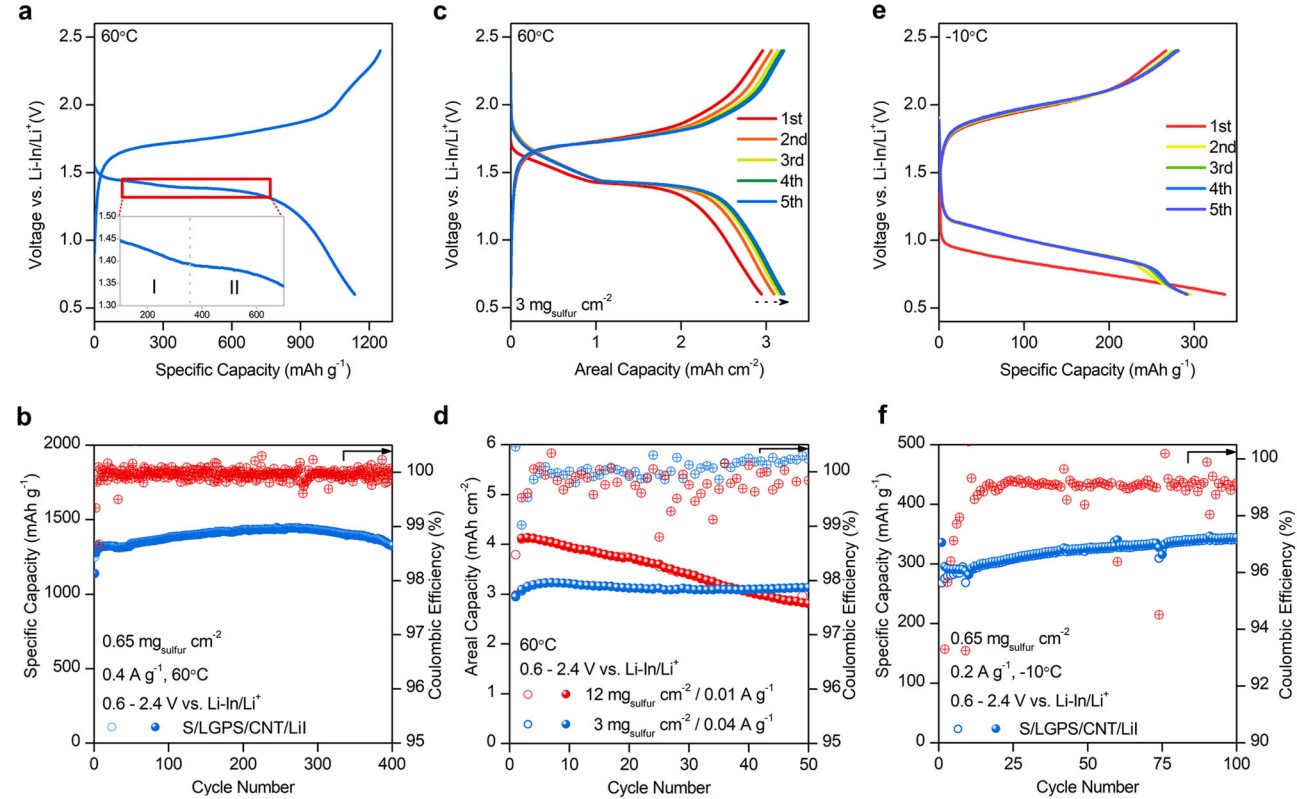

**Fig. 4 | Electrochemical behavior of all-solid-state Li-S batteries under various operating temperatures. a** Voltage profile of ASSLSB tested at 60 °C, with enlarged discharge profile shown between 1.3–1.5 V (vs. Li-In/Li+). **b** Cycling performance of ASSLSB at 0.4 A g−1 and 60 °C. **c** Voltage profile of ASSLSB with active material loading of 3 mg cm−2. **d** Cycling performance of high loading ASSLSBs at 60 °C. **e** Voltage profile of ASSLSB tested at 0.2 A g−1 and −10 °C. **f** Cycling performance of ASSLSBs at 0.2 A g−1 and −10 °C.

50 cycles (Fig. 4d). The cycling stability of the 12 mg cm$^{-2}$ loading cell is poor compared to the 3 mg cm$^{-2}$ cell, which is likely due to severe volume change of sulfur during (de)lithiation. Volume change induces contact loss between the active material, SSE, carbon, which increases internal cell resistance, and considerably limits cycling stability as a result.

Figure 4e shows the voltage profile of the LiI-incorporated cell tested at −10 °C. The overpotential increases considerably, which can be attributed to slow charge transfer kinetics within the S composite electrode at low temperature. Still, the cell is fully reversible and exhibits a relatively high initial discharge capacity of 336 mAh g$^{-1}$, maintaining a stable capacity for over 100 cycles (Fig. 4f). These results demonstrate the effectiveness of the catalytic incorporation of LiI for achieving fully reversible all-climate ASSLSBs with high active material loading.

Understanding the discharge products of electrochemical energy storage systems such as metal-air and metal-sulfur batteries has proven crucial for enhancing key performance metrics such as active material utilization, specific capacity, cycle life, and reversibility[66–69]. In this regard, elucidating the intricate relationship between the discharge products and the electrochemical behavior of ASSLSBs is critical but remains inadequately studied thus far. In this study, X-ray absorption spectroscopy was employed to reveal that the discharge product of ASSLSBs is not exclusively composed of $Li_2S$ but rather a mixture of $Li_2S$ and $Li_2S_2$. Time-of-flight secondary ion mass spectrometry was utilized to validate the presence of $Li_2S_2$ by detecting its characteristic ion, $Li_3S^{2+}$, and to quantify the relative proportion of $Li_2S_2$ in the $Li_2S$ and $Li_2S_2$ mixture using the ion intensity ratio of $Li_3S^{2+}/Li_3S^+$. Density functional theory calculations were employed to showcase that while $Li_2S_2$ exhibits superior redox kinetics compared to $Li_2S$, both species hinder the reversibility of ASSLSBs. Building upon these findings, an integrated strategy was proposed to enhance the reversibility and cycling stability of ASSLSBs. This approach involved: (1) manipulating the lower cutoff potential of ASSLSBs to promote the formation of $Li_2S_2$-dominant discharge product and (2) incorporating a trace amount of LiI into the S composite electrode to improve the electrochemical oxidation of $Li_2S_2$ and $Li_2S$. As a result, ASSLSBs delivered a reversible capacity of 979.6 mAh g$^{-1}$ for 1500 cycles at 2.0 A g$^{-1}$ at 25 °C and demonstrated stable cycling stability across a wide temperature range (−10, 25, and 60 °C). Furthermore, high active material loading ASSLSBs were tested and achieved areal capacities exceeding 3.0 mAh cm$^{-2}$, demonstrating the practical viability of this approach. In summary, this work utilizes advanced analytical techniques to probe the discharge products of ASSLSBs, yielding valuable insights into their electrochemical behavior and resulting in strategies that can be widely adopted to achieve fully reversible, all-climate ASSLSBs with high capacity, long lifetime, and enhanced safety.

## Methods

### Preparation of sulfur composite electrodes
A mixture of sulfur powder (Sigma-Aldrich), $Li_{10}GeP_2S_{12}$ (MSE supplies), and carbon nanotubes (Sigma-Aldrich) with a weight ratio of 36:40:24 was transferred into a 50 mL agate ball-milling jar filled with 40 g of 5 mm zirconia balls under an Ar atmosphere ($H_2O$ < 0.1 ppm, $O_2$ < 0.1 ppm). The mixture was ball-milled using a high-speed ball-milling machine at 200 rpm for 4 h. The same procedure was used to prepare the LiI-incorporated S composite electrodes.

### Materials characterization
Powder X-ray diffraction (XRD) patterns were recorded on a Bruker AXS D8 Advance instrument with Cu Kα radiation (λ = 1.5406 Å). The sample holder was covered with Kapton tape to prevent air exposure. Raman spectra were obtained on a HORIBA Scientific LabRAM HR Raman spectrometer system (532.4 nm laser). Thermogravimetric analysis was performed using a thermal analyzer (Diamond TG,

PerkinElmer, USA) under a nitrogen atmosphere using a heating rate of 5 °C min$^{-1}$. Scanning electron microscopy (SEM) images were recorded using a FE-SEM (S4800, Hitachi high-technologies) equipped with an energy-dispersive X-ray spectroscopy (EDS) system. XAS was carried out at the Canadian Light Source (CLS). Sulfur $K$-edge XAS was collected using total electron yield (TEY) mode on the Soft X-ray Microcharacterization beamline (SXRBM) at the CLS. To achieve a good signal to noise ratio, an ambient table setup was used at the SXRMB beamline. The chamber was filled with helium gas to reduce absorption and scattering at low energies. The S composite electrodes and pure $Li_2S$ powder pressed on an aluminum foil were examined using TOF-SIMS IV (ION-TOF GmbH, Germany) equipped with a BiMn cluster liquid metal ion source. A pulsed 25 keV $Bi_3^+$ primary cluster ion beam was used to generate secondary ions from the topmost 1–3 nm of the sample surface. Ion mass spectra, i.e., intensities of ions against mass to charge ratio (m/z), were collected at three spots in an area of 200 × 200 µm$^2$. A pulsed, low energy electron flood was used to neutralize the sample so that insulating samples can be measured. Positive secondary ion mass spectra were calibrated by $Li^+$, $CH_3^+$ and $C_3H_5^+$, while negative ones by $Li^-$, $CH^-$ and $S^-$. The mass resolutions of $CH_3^+$ and $C_3H_5^+$ were 4000 and 5200, respectively, while the mass resolutions of CH and $C_2H$ were 3400 and 4000, respectively. For comparison purposes, the spectra shown in Supplementary Fig 10 were normalized to their total ion intensities. X-ray photoelectron spectroscopy (XPS) testing was conducted using a monochromatic Al Kα source (1486.6 eV) in a Kratos AXIS Nova Spectrometer. The Ar-filled glovebox was connected to the XPS machine to avoid exposure to air.

### First principles calculations
All the first principles calculations were carried out in the DFT framework implemented in the VASP package[70]. The projector augmented-wave pseudopotentials were used to describe the interaction between ions and electrons, and the exchange-correlation effects were treated using the Perdew–Burke–Ernzerhof (PBE) functional under the generalized gradient approximation (GGA)[71]. A Monkhorst–Pack k-point grid of 3 × 3 × 1, and a kinetic energy cut-off of 600 eV was used to optimize all surface calculations. The LiI surface was created from a 3 × 3 × 3 supercell and a vacuum of 15 Å was used to avoid interaction between images. For surface calculations, the van der Waals (vdW) correction function proposed by Grimme was utilized[72]. All the atoms were optimized until the total energies converged to below 10$^{-4}$ eV and the forces acting on atoms were less than 10$^{-2}$ eV/Å. The adsorption energy ($E_A$) was calculated using the expression $E_A = E_{surface+adsorbate} - (E_{surface} + E_{adsorbate})$. The formation energy ($E_f$) was calculated using the expression $E_f = E_{xy} - (E_x + E_y)$, where x and y are pristine elements forming compound xy. The structures were visualized using the VESTA package[73].

### Electrochemical testing/characterization
ASSLSBs were assembled inside an Ar-filled glovebox and tested using model cells. First, 120 mg of $Li_{10}GeP_2S_{12}$ was placed into a polytetra-fluoroethylene (PTFE) die with a diameter of 10 mm and pressed at 1 ton. The thickness of the SSE layer was approximately 1 mm. Next, approximately 1.5 ~ 2 mg of S composite electrode powder was dispersed onto the $Li_{10}GeP_2S_{12}$ side and pressed at 3 tons. The mass loading of S was approximately 0.65 mg cm$^{-2}$. Finally, a Li-In alloy was placed on the bare $Li_{10}GeP_2S_{12}$ side and pressed at 1 ton. The Li-In alloy was prepared by pressing a piece of In foil (ɸ 10 mm, thickness 0.1 mm) and a piece of Li foil (ɸ 10 mm, thickness 20 µm) together under ~60 MPa for 5 min. All the batteries were tested under an external pressure of ~150 MPa. ASSLSBs were tested within the voltage range of 0.6–2.4 V (vs. Li-In/Li$^+$) using a Land cycler (Wuhan, China). Battery testing at 25 °C was conducted in a designated battery testing lab equipped with a temperature control system to ensure accurate temperature conditions. For testing at −10 °C, a freezer manufactured by

Thermo Fisher Scientific was used to create the desired low temperature environment. For testing at 60 °C, a convection oven manufactured by Thermo Fisher Scientific was employed to achieve the required high temperature conditions. All cells underwent a resting period and were allowed to equilibrate for 12 h to ensure that they reached the target temperatures and stabilized before the actual testing took place. The galvanostatic intermittent titration technique (GITT) was performed using a constant specific current of 0.2 A g$^{-1}$ for 20 min followed by a relaxation period of 2 h during the charge/discharge process in the first cycle.

## Data availability
The datasets generated during and/or analyzed during the current study are available from the corresponding author on request.

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

## Acknowledgements

This work was supported by Natural Sciences and Engineering Research Council of Canada (NSERC); Canada Research Chair Program (CRC); Canada Foundation for Innovation (CFI); Ontario Research Fund (ORF); Canada Light Source (CLS) at University of Saskatchewan; Inter-disciplinary Development Initiatives (IDI) by Western University; and University of Western Ontario. C.W. acknowledges the Banting Post-doctoral Fellowship (BPF – 180162). J.T.K. thanks Professor Yang Zhao and Professor Yukwon Jeon for their thought-provoking discussions.

## Author contributions

J.T.K. and C.W. conceived the study and analyzed the results. J.T.K. conducted the experiments and collected data. J.T.K. wrote the manuscript. A.R. conducted the DFT calculations. S.D. helped with XANES measurements. W.L. and F.Z. helped with XRD and SEM characterization. J.L. and H.D. helped with Raman characterization. H-Y. N. helped with ToF-SIMS characterization. Y.H., J.F., and X.H. participated in the discussion of the data. X.S., C.V.S., and C.W. supervised the project. All authors have approved the final version of the manuscript.

## Competing interests

The authors declare no competing interests.
