## [Peer Review File · Nature Communications]

Manipulating Li₂S₂/Li₂S Mixed Discharge Products of All-Solid-State Lithium Sulfur Batteries for Improved Cycle LifeREVIEWER COMMENTS

Reviewer #1 (Remarks to the Author):

Review comments

Ms. No.: NCOMMS-23-04993-T

Title: Revealing Li₂S₂/Li₂S Mixed Discharge Products of All-Solid-State Li-S Batteries for Ultralong Cycling Life

In this work, the authors deal with the discharge products of Li-S batteries. The area of research Li-S battery as an emerging technology seeks ample attention. The two most important aspects that the authors want to project in this study are: 1) to demonstrate that the discharge products of Li-S batteries include Li₂S₂ along with Li₂S, 2) the Li₂S₂ is electrochemically more sustainable than the Li₂S which can be utilized for the designing future Li-S configurations. X-ray absorption spectroscopy and time-of-flight secondary ion mass spectrometry have been employed to reveal the mixed phase of the discharge products. The presence of Li₂S₂ during the discharge of Li-S configurations has been reported in some previous attempts. The addition of LiI to improve the oxidation characteristics has also been proved in some of the previous attempts in Li-S configurations. Even though the research direction and the research problem are remarkably imperative, numerous flaws make the manuscript incompatible with a wider audience of the Journal. Some important remarks are:

1. The style of presentation of the results in the manuscript has to be further improved.
2. The Authors have to clearly mention the precedents on the major milestones to probe out the discharge products of Li-S batteries, pointing to the novelty of the present attempt. In the introduction part, the authors have to explain the state-of-the-art research work to reveal the discharge products of Li-S configurations, indicating the novelty of their attempt beyond the state-of-the-art research.
3. The authors' mentioned that "In this study, we first interpret the active material utilization of ASSLSBs reported in 'recent literature' to postulate premise of a mixed discharge product as a result." Here, the authors didn't cite any specific literature for analysis to reach the postulate.
4. The polysulfide peak has been identified in some earlier attempts indicating the presence of Li₂S₂ as the discharge product using the X-ray photoelectron spectroscopy. In that case, how they can claim the mixed discharge product has been identified as the first of its kind? Again, the pre-edge feature appears at 2471.3 eV in the XANES study and is not much prominent for unambiguous identification of the Li₂S₂ phase during discharge. The authors have to comment on this in the result and discussion part.
5. The authors' attempted to improve the Li₂S₂ dominant discharge products during cycling by decreasing the cut-off voltage without providing an explanation on how they came to the conclusion that more Li₂S₂ will be resulted by decreasing the cut-off voltage. More explanations are needed.
6. The authors state: "Along with low discharge capacity, the existence of Li₂S₂ can help (to) explain another feature of ASSLSBs that is prevalent across literature: poor electrochemical reversibility after the initial discharge cycle." The authors claim that Li₂S₂ boosts the electrochemical redox activities better than Li₂S. However, the above sentence is contradictory to their claim. They have to provide a proper explanation of this. Again in XAS, the authors mentioned that "After the charge, the Li₂S and Li₂S₂ features become weaker but are still present in the spectra, which indicates the irreversible transformation from Li₂S₂/Li₂S to S" In that case how the mixed discharge product with dominant Li₂S₂ phase improves the long term cyclability?
7. It is well known that Li₂S₂ exhibits a metastable phase and spontaneously dissociates to Li₂S and S. The authors have to clearly explain the possible overall redox reactions that occur in the Li-S configurations when Li₂S₂ mixed discharge products exist. Again, the authors have to give a proper explanation of the stability of the Li₂S₂ phase as a discharge product in Li-S.
8. The addition of LiI to promote the oxidation process in Li-S batteries has been reported earlier. When the mixed discharge product with excess Li₂S₂ is formed, how does the LiI contribute to promoting the redox activities, than in the case of the formation of a single discharge product (Li₂S)?

How the authors fixed the optimum quantity of LiI added to improve the oxidation? If they followed some previous attempts, the authors have to mention that.

9. In the rate performance of the Li-S configuration with excess Li_2S_2 component, the initial capacity of the sample at 0.2 A g^{-1} is 933 mAh g^{-1} , which is improved at higher current densities. The authors provided the explanation for the gradual improvement in improvement in the capacity of the sample with the aid of two processes (1) Initial sulfur activation; 2) The electrochemically active discharge product of LGPS solid electrolyte). However, after reaching the current rate of 6.0 A g^{-1} (303.7 mAh g^{-1}), the recovering capacity at 0.2 A g^{-1} has been reported as $1222.4 \text{ mAh g}^{-1}$. Which is much higher than all other obtained capacities. The aforementioned two explanations are not sufficient to explain this high-recovered capacity obtained in the sample since the sulfur activation and SE dissociation are the initial processes. The authors have to give some explanations for this.

10. The overall style of presentation and the many sentences in the manuscript has to be further modified.

The authors have to address the aforementioned points before the consideration of the manuscript for publication.

Reviewer #2 (Remarks to the Author):

All-solid-state lithium-sulfur (Li-S) batteries are regarded as the most promising energy storage system because of their low cost and theoretically high energy density. However, the mechanism of all-solid-state Li-S batteries remains elusive. The study by Kim et al. investigates the root cause of the lower practical discharge capacities of all-solid-state Li-S batteries than their theoretical value. By virtue of advanced characterization techniques such as XAS and ToF-SIMS, the authors identified that the mixed discharge product of $\text{Li}_2\text{S}_2/\text{Li}_2\text{S}$ accounts for the lower practical discharge capacities. DFT calculations further confirms that Li_2S_2 has better electrochemical kinetics than Li_2S . Based on this understanding, the authors devise a strategy that involves modifying the lower cut-off potential to promote a Li_2S_2 -dominant discharge product and incorporating LiI into the cathode composite. This approach results in fully reversible ASSLSBs with outstanding cycling stability for over 1500 cycles at room temperature. In summary, this work represents a significant advancement in ASSLSB research by offering novel insights into the discharge products of ASSLSBs and demonstrating unprecedented cycling results. The strategy presented in this study is highly practical. Therefore, I would like to recommend it for publication in the prominent journal Nature Communications. Also, there are some minor comments, as follows:

1. The authors propose that by manipulating the lower cut-off voltage and inducing a discharge product dominated by Li_2S_2 , the reversibility and cycling stability of ASSLSBs can be enhanced. I suggest authors to provide more data with different lower cut-off voltages to strengthen their claim.
2. Why did the authors specifically choose 6 wt% LiI? The effect of LiI content should be investigated.
3. Does LiI contribute any capacity during the charge/discharge process?
4. What are the reasons for the large electrochemical polarization in discharge curves at low temperature in Figure 4e?
5. Why are there two plateaus in the discharge curves when testing 60°C ?
6. Typo in figure 2 caption 'cia'
7. There are two figure 4's in this manuscript – the latter should be figure 5

Point-by-Point Response to Referee's Comments

(NCOMMS-23-04993-T)

Reviewer #1 (Remarks to the Author):

Review comments

Ms. No.: NCOMMS-23-04993-T

Title: Revealing Li₂S₂/Li₂S Mixed Discharge Products of All-Solid-State Li-S Batteries for Ultralong Cycling Life.

In this work, the authors deal with the discharge products of Li-S batteries. The area of research Li-S battery as an emerging technology seeks ample attention. The two most important aspects that the authors want to project in this study are: 1) to demonstrate that the discharge products of Li-S batteries include Li₂S₂ along with Li₂S, 2) the Li₂S₂ is electrochemically more sustainable than the Li₂S which can be utilized for the designing future Li-S configurations. X-ray absorption spectroscopy and time-of-flight secondary ion mass spectrometry have been employed to reveal the mixed phase of the discharge products. The presence of Li₂S₂ during the discharge of Li-S configurations has been reported in some previous attempts. The addition of LiI to improve the oxidation characteristics has also been proved in some of the previous attempts in Li-S configurations. Even though the research direction and the research problem are remarkably imperative, numerous flaws make the manuscript incompatible with a wider audience of the Journal. Some important remarks are:

Response: We are grateful to the reviewer for their thorough evaluation of our manuscript. We highly appreciate the reviewer's constructive and valuable feedback, and we have carefully considered these suggestions in our response.

1. The style of presentation of the results in the manuscript has to be further improved.

Response: We sincerely value the reviewer's valuable feedback, and we have dedicated considerable efforts to this revision, diligently addressing all their concerns

and incorporating their suggestions. As a result, we have successfully enhanced the quality of our manuscript, improved its overall flow, and effectively presented the results.

2. The Authors have to clearly mention the precedents on the major milestones to probe out the discharge products of Li-S batteries, pointing to the novelty of the present attempt. In the introduction part, the authors have to explain the state-of-the-art research work to reveal the discharge products of Li-S configurations, indicating the novelty of their attempt beyond the state-of-the-art research.

Response: We appreciate the valuable feedback provided by the reviewer regarding our manuscript. We agree with the reviewer that we have not adequately discussed prior work in relation to probing the discharge products of all-solid-state Li-S batteries. Furthermore, we agree that we need to emphasize the originality of our study in comparison to the existing research. In this context, it is important to note that our study is built upon a limited body of previous studies that have specifically focused on examining the charge/discharge mechanism and discharge products of all-solid-state Li-S batteries. We have compiled a list of the three previous works that are closely related to our research topic and summarized them for the reviewer's reference.

(1) The first study (Adv. Energy Mater. 2016, 6, 1600806) investigates the electrochemical lithiation process from S_8 to Li_2S by conducting an in-situ transmission electron microscopy (TEM) study on carbon-coated sulfur materials in all-solid-state Li-S batteries. A three-step lithiation process of S_8 was identified. First, S_8 particles undergo a partial transformation into nanocrystalline S and nanocrystalline Li_2S . Second, the Li_2S nanocrystals gradually grow larger as the lithiation process proceeds, reducing the diffusion distance for electrons and Li^+ ions at the S/ Li_2S interfaces. Third, as the S phase decreases, the Li_2S phase increases, resulting in the formation of pure Li_2S at the end of the lithiation process. It is important to highlight that this study reports the direct transformation of S into Li_2S without the formation of intermediate lithium polysulfide species (Li_2S_x , $4 \leq x \leq 8$). The conclusion of this work diverges from

the findings and viewpoints stated in our manuscript, where we demonstrate the discharge product of all-solid-state Li-S batteries is a mixture of Li_2S_2 and Li_2S .

- (2) The second study (Small 2020, 16, 2001899) employs an in-situ TEM technique implemented with a microelectromechanical systems (MEMS) heating device to study the precipitation and decomposition of Li_2S at high temperatures. It was revealed that the decomposition of Li_2S , an electrochemical process that is typically hindered at ambient temperatures, becomes feasible at higher temperatures. Because Li^+ ion diffusion has stronger temperature-dependence than electron conductivity, these results suggest that the reversibility of Li_2S in solid-state LSBs is governed by the Li^+ ion diffusion energy rather than electronic conductivity.
- (3) The third study that was just recently published (Angew. Chem.Int. Ed. 2023, 62, e2023023) investigates the electrochemical reaction pathway of all-solid-state Li-S batteries, aiming to elucidate the differences when compared to their liquid Li-S counterparts. Notably, the study reveals that the electrochemical redox pathway within all-solid-state Li-S batteries follow a solid-solid conversion from S_8 to Li_2S , involving the formation of an intermediate Li_2S_2 phase. To comprehensively examine the Li_2S_2 phase, characterization techniques such as XANES and first-principles calculations are employed. Our study corroborates these findings and adds significant value in several key areas. First, we interpret the active material utilization of ASSLSBs reported in recent literature to postulate a mixed discharge product consisting of Li_2S and Li_2S_2 . Second, along with XANES characterization, we showcase the utilization of ToF-SIMS, a cutting-edge analytical technique, to directly determine the chemical composition of Li_2S_2 and identify that the final discharge product of all-solid-state Li-S batteries is a mixture of Li_2S_2 and Li_2S . Third, we thoroughly investigated the correlation between the discharge product ($\text{Li}_2\text{S}_2/\text{Li}_2\text{S}$) and the electrochemical behavior of all-solid-state Li-S batteries, encompassing crucial aspects such as initial discharge capacity, cycling stability, and electrochemical reversibility. Finally, we identify a promising strategy to enhance the electrochemical performance of all-solid-state Li-S

batteries by utilizing insights gained from the behavior and characteristics of Li_2S_2 .

To enhance the coherence and fluency of our manuscript, we added a new paragraph to the introduction of the revised manuscript that highlights major milestones to investigate the charge/discharge mechanism of all-solid-state Li-S batteries.

REVISION (Page 3): "While these strategies have proven fruitful, a key obstacle hindering the development of ASSLSBs stems from conceptual ambiguity surrounding their underlying redox mechanisms. Initial research employing in situ transmission electron microscopy explored the evolution of Li_2S in ASSLSBs, revealing a three-step lithiation process and direct conversion from S_8 to Li_2S , without the formation of other sulfur species.²¹ Another study investigated the decomposition behavior of Li_2S highlighting that the decomposition of Li_2S is governed by Li^+ ion conductivity rather than electronic conductivity.²² A recent study investigating the electrochemical reaction pathway of ASSLSBs reported the presence of a Li_2S_2 intermediate phase during the conversion from S_8 to Li_2S .²³ These studies have set important precedents and resulted in a richer understanding of the fundamental redox mechanisms of ASSLSBs. However, the intricate interplay between the discharge products and the electrochemical behavior of ASSLSBs, encompassing crucial aspects such as initial discharge capacity, cycling stability, and reversibility, remains insufficiently explored but stands as a pivotal prerequisite for driving the advancement of ASSLSB technology."

3. The authors' mentioned that "In this study, we first interpret the active material utilization of ASSLSBs reported in 'recent literature' to postulate premise of a mixed discharge product as a result." Here, the authors didn't cite any specific literature for analysis to reach the postulate.

Response: We thank the reviewer for bringing this issue to our attention. We have included citations for the publications we analyzed in this study to reach our postulate in the introduction of the revised manuscript. The **first paragraph** of the results section

and **Figure 1b** in the main manuscript further highlights the studies we used for analysis in this work and how we reached the premise of a mixed discharge product.

REVISION (Page 4): In this study, we first interpret the active material utilization of ASSLSBs reported in recent literature^{20,24–32} to postulate a mixed discharge product consisting of lithium sulfide (Li_2S) and lithium disulfide (Li_2S_2).

Figure 1: (b) Initial discharge capacities of ASSLSBs recently reported in literature.

4. The polysulfide peak has been identified in some earlier attempts indicating the presence of Li_2S_2 as the discharge product using the X-ray photoelectron spectroscopy. In that case, how they can claim the mixed discharge product has been identified as the first of its kind? Again, the pre-edge feature appears at 2471.3 eV in the XANES study and is not much prominent for unambiguous identification of the Li_2S_2 phase during discharge. The authors have to comment on this in the result and discussion part.

Response: We appreciate the reviewer's insightful question. For all-solid-state Li-S batteries, the common understanding is that S_8 directly converts to Li_2S (*Nano Lett.* 2019, 19, 5, 3280–3287; *Adv. Energy Mater.* 2016, 6, 1600806; *Small* 2022, 18, 2106970). Therefore, previous studies have not used XPS to identify Li_2S_2 in all-solid-state Li-S batteries, even though XPS has been utilized previously to analyze Li_2S_2 in liquid Li-S batteries (*Sci Rep* 5, 12146 (2015); *Nat Commun* 6, 5682 (2015)). The pre-edge feature that appears at 2471.3 eV in the XANES results provides some evidence

of the Li_2S_2 phase during discharge. However, we agree with the reviewer that the XANES results alone are not enough to unambiguously identify Li_2S_2 . Therefore, we employed XPS to investigate the presence of Li_2S_2 in solid-state Li-S batteries. However, as illustrated in **Figure R1**, we did not observe a discernible Li_2S_2 peak in the XPS spectra. Previous studies that investigate all-solid-state Li-S batteries using XPS show similar results, where no distinct Li_2S_2 peak is evident (*Chem. Mater.* 2019, 31, 8, 2930–2940; *Nano Energy*, 96 (2022), Article 107093). We speculate that the chemical similarity and overlapping peaks of Li_2S_2 and Li_2S , along with the challenges in isolating Li_2S_2 as a reference sample, limit the accuracy of traditional XPS analysis in identifying Li_2S_2 .

Figure R1. S 2p XPS Spectra of ASSLSBs (a) before cycling, (b) after full discharge, and (c) after full charge. The S $2p_{3/2}$ binding energies of S_0 , PS_4^{3-} , and $\text{S}^{2-}/\text{Li}_2\text{S}$, are 163.3 eV, 161.3 eV, and 1601.1 eV, respectively.

To support the XANES results, we utilized advanced characterization techniques to successfully identify the Li_2S_2 phase in solid-state Li-S batteries. Specifically, we introduced ToF-SIMS, which demonstrates superior chemical selectivity compared to XPS, enabling effective differentiation between Li_2S and Li_2S_2 . Our study represents a pioneering application of ToF-SIMS for investigating the discharge products of solid-state Li-S batteries. Prior to our work, the ToF-SIMS technique had not been employed for such purposes, even to study the discharge products of liquid Li-S batteries, underscoring the novelty and significance of our approach.

To enhance the clarity and novelty of our findings, we have included **Figure R1** as **supplementary figure 9** and incorporated an additional paragraph in the results section.

REVISION (Page 12): "As for chemical analyses of Li_2S_2 and Li_2S , X-ray photoelectron spectroscopy (XPS) has been used previously to investigate the chemical composition of Li_2S_2 and Li_2S in liquid Li-S batteries.^{59,60} The detection of a S $2p_{3/2}$ peak at 162.2 eV is attributed to Li_2S_2 due to its binding energy's proximity to the reference sample of Na_2S_2 (162.0 eV). We utilized XPS to complement the XANES results and confirm the presence of the Li_2S_2 phase. However, as illustrated in **Supplementary Fig. 9**, a discernible Li_2S_2 peak was not found. Previous studies that investigate ASSLSBs using XPS show similar results, where no distinct Li_2S_2 peak is evident in the XPS spectra.^{32,61} Chemical similarity and overlapping peaks of Li_2S_2 and Li_2S pose challenges in accurately identifying Li_2S_2 using traditional XPS analysis. Additionally, the difficulty in isolating Li_2S_2 as a reference sample further complicates the analysis.

To directly determine the chemical identity of Li_2S_2 and gather additional supporting evidence for its existence, we used ToF-SIMS, which demonstrates superior chemical selectivity compared to XPS, enabling effective differentiation between Li_2S and Li_2S_2 ."

5. The authors' attempted to improve the Li_2S_2 dominant discharge products during cycling by decreasing the cut-off voltage without providing an explanation on how they came to the conclusion that more Li_2S_2 will be resulted by decreasing the cut-off voltage. More explanations are needed.

Response: We thank the reviewer for bringing this ambiguity to our attention. To elucidate the basis for our conclusion, we conducted ToF-SIMS analysis on two batteries: one discharged to 1.2V (vs. Li^+/Li) and another discharged to 0.4V (vs. Li^+/Li), as shown in **Figure R2** below.

Figure R2. Ion intensity of Li₃S⁺ for ASSLSBs discharged to 1.2V (vs. Li⁺/Li) and 0.4V (vs. Li⁺/Li). The Li₃S⁺ ion intensity is normalized to the total ion intensity. The error bars represent the standard deviation of the measured intensity.

The normalized peak intensity of Li₃S⁺ ions (formed from Li₂S) for the battery discharged to 1.2V is considerably lower compared to the battery discharged to 0.4V. This suggests that the battery discharged to 1.2V exhibits a higher proportion of Li₂S₂ and a correspondingly lower proportion of Li₂S, when contrasted with the battery discharged to 0.4V.

To enhance the clarity of our methodology for promoting a Li₂S₂-dominant discharge product, we have incorporated **Figure R2** as **supplementary figure 11** and included further discussion in the revised manuscript.

REVISION (Page 12): “ToF-SIMS analysis reveals that the normalized intensity of Li₃S⁺ ions decreased considerably in the cell discharged to 1.2 V compared to the one discharged to 0.4 V, which suggests that a Li₂S₂-dominant discharge product can be obtained by limiting the lower voltage threshold (**Supplementary Fig. 11**).”

6. The authors state: “Along with low discharge capacity, the existence of Li₂S₂ can help (to) explain another feature of ASSLSBs that is prevalent across literature: poor electrochemical reversibility after the initial discharge cycle.” The authors claim that Li₂S₂ boosts the electrochemical redox activities better than Li₂S. However, the above

sentence is contradictory to their claim. They have to provide a proper explanation of this. Again in XAS, the authors mentioned that “After the charge, the Li₂S and Li₂S₂ features become weaker but are still present in the spectra, which indicates the irreversible transformation from Li₂S₂/Li₂S to S” In that case how the mixed discharge product with dominant Li₂S₂ phase improves the long term cyclability?

Response: We thank the reviewer for bringing this to our attention. We have recognized the need for clarity in the presentation of the paragraph that the reviewer refers to, as it may easily perplex the reader. The objective of this paragraph is to convey that the irreversibility of all-solid-state Li-S batteries after the initial discharge cycle has previously been attributed to the irreversible formation of Li₂S. However, the impact of Li₂S₂ on the reversible capacity of all-solid-state Li-S batteries has not been thoroughly examined. To address this gap, we conducted density functional theory (DFT) calculations for both Li₂S₂ and Li₂S and discovered that Li₂S₂ exhibits superior redox activity compared to Li₂S. However, it is important to note that a Li₂S₂-dominant discharge product does not guarantee fully reversible ASSLSBs. While the formation energy of Li₂S₂ (-1.01 eV/atom) is higher than that of Li₂S (-1.59 eV/atom), it remains significantly lower than that of S₈. Consequently, it is likely that both Li₂S₂ and Li₂S contribute to the irreversible capacity of all-solid-state Li-S batteries. Prior research has shown that the addition of Lil can enhance the redox activity of Li₂S, resulting in fully reversible all-solid-state Li-S batteries (*ACS Appl. Energy Mater.* 2022, 5, 8, 9429–9436; *Nano Lett.* 2021, 21, 19, 8488–8494; *Nat Commun* 12, 5943 (2021)). In our investigation, we conducted additional DFT calculations and observed that Lil can reduce the activation energy for both Li₂S₂ and Li₂S, thereby facilitating their electrochemical oxidation back to S₈. Hence, we introduced a trace quantity of Lil to enable the full electrochemical oxidation of both Li₂S₂ and Li₂S.

We have completely re-written **paragraph 3** of the results section to provide more clarity regarding the redox activities of Li₂S₂/Li₂S and its influence on the reversibility of all-solid-state Li-S batteries.

REVISION (Page 7): “Another recurrent feature observed in the literature regarding ASSLSBs is their poor electrochemical reversibility, particularly following the initial

discharge cycle.^{15,20,41} This phenomenon has been attributed to the irreversible formation of Li_2S , where the stable antifluorite structure of Li_2S necessitates high activation potentials, typically approaching 4V (versus Li^+/Li), to facilitate the electrochemical oxidation (or delithiation) of Li_2S back to S_8 during the charging process.^{42–45} We conducted density functional theory (DFT) calculations to investigate the influence of Li_2S_2 and Li_2S on the reversibility of ASSLSBs (**Supplementary Note 1**). The calculated formation energies of Li_2S_2 and Li_2S were approximately -1.01 eV/atom and -1.59 eV/atom, respectively (**Fig. 1d**). These results indicate that Li_2S_2 exhibits better redox activities compared to Li_2S . The formation energy of Li_2S_2 , however, remains considerably lower than that of S_8 . This suggests that both Li_2S_2 and Li_2S hinder the electrochemical reversibility of ASSLSBs. Previous studies have demonstrated the effective use of lithium iodide (LiI) to enhance the electrochemical oxidation of Li_2S , thereby achieving fully reversible ASSLSBs.^{16,46,47} Indeed, our DFT calculations reveal that the molecular conversion of $\text{Li}_2\text{S}_2/\text{Li}_2\text{S}$ to S_8 on the LiI(100) surface requires a lower activation barrier compared to the process in vacuum (**Fig. 1d, e**). These results suggest that LiI can facilitate the electrochemical oxidation of not only Li_2S but also Li_2S_2 , thereby improving the reversibility of ASSLSBs as a result. Further discussion regarding the DFT calculations and the role of LiI in promoting the electrochemical oxidation of $\text{Li}_2\text{S}_2/\text{Li}_2\text{S}$ is provided in **Supplementary Note 1.**"

To determine how the mixed discharge product with a dominant Li_2S_2 phase improves long term cyclability, we assembled two ASSLSBs and tested them using a lower cut-off voltage of 1.2V and 0.4V (vs. Li^+/Li) for comparison. The voltage profile and cycling data of the ASSLSBs are shown in Figure R3 below.

Figure R3. (a) Voltage profile of ASSLSBs tested using different lower limit potentials. (b) Corresponding cycling performance of ASSLSBs tested at 0.2 A g⁻¹ and 25°C.

Lowering the battery's potential to 0.4 V enhances its initial discharge capacity to approximately 1300 mAh g⁻¹, compared to around 850 mAh g⁻¹ for the battery discharged to 1.2 V (**Figure R3a**). The higher initial discharge capacity of the 0.4 V cell can be attributed to an increased formation of Li₂S. Furthermore, both cells contain 6 wt% LiI, however, only the 1.2 V discharged cell is fully reversible on charge, while the 0.4 V discharged cell only reaches about 80% of charge capacity. This suggests that more than 6 wt% LiI is needed to electrochemically address the extra Li₂S that is formed.

A mixed discharge product with a dominant Li₂S₂ phase enhances long-term cycling stability due to two main reasons. First, Li₂S₂ exhibits better kinetics compared to Li₂S, as supported by our DFT results. This improved kinetics leads to enhanced reversibility, contributing to better cycling performance. Second, the volume expansion of Li₂S₂ is lower than that of Li₂S, with Li₂S₂ expanding by approximately 60% compared to Li₂S's expansion of around 80%. This reduced volume expansion helps alleviate the volume-induced failures that can occur during cycling. **Figure R3b** illustrates the significant improvement in cycling stability observed in the cell with a Li₂S₂-dominant discharge product.

To provide further clarification on the advantages of a Li_2S_2 -dominant discharge product, we have included **Figure R3** as **supplementary figure 12** and expanded the discussion in the revised manuscript.

REVISION (Page 13): "Although promoting a discharge product with an Li_2S_2 -dominant phase comes at the expense of the initial discharge capacity, doing so enhances both the reversibility and cycling stability of ASSLSBs (**Supplementary Fig. 12**). This is because Li_2S_2 is more electrochemically active than Li_2S , and the volumetric expansion of Li_2S_2 is comparatively smaller, at ~60%, compared to Li_2S which expands by approximately 78%."

7. It is well known that Li_2S_2 exhibits a metastable phase and spontaneously dissociates to Li_2S and S. The authors have to clearly explain the possible overall redox reactions that occur in the Li-S configurations when Li_2S_2 mixed discharge products exist. Again, the authors have to give a proper explanation of the stability of the Li_2S_2 phase as a discharge product in Li-S.

Response: We thank the reviewer for their constructive comments. Indeed, first principles calculations have demonstrated the existence of a metastable phase of Li_2S_2 , wherein spontaneous disproportionation into Li_2S and S occurs to minimize the Gibbs free energy of the system (*J. Mater. Chem. A*, 2015, **3**, 8865-8869; *Phys. Chem. C* 2015, **119**, **9**, 4675–4683). However, it is worth noting that these calculations typically focus on the bulk Gibbs free energy, overlooking the significant contribution of the surface Gibbs free energy, particularly in the case of nanosized particles. Previous studies suggest that a large fraction of Li_2S_2 can remain stabilized due to the influence of surface Gibbs free energy (*Phys. Chem. Lett.* 2017, **8**, **7**, 1324–1330; *Nano Lett.* 2014, **14**, **2**, 1016–1020). Moreover, it has been reported that Li-S batteries operate under non-equilibrium thermodynamic conditions during the charge/discharge process, which inherently promotes the formation of metastable Li_2S_2 (*J. Power Sources*, 272 (2014), pp. 518-521). The possible overall redox reactions that occur in all-solid-state Li-S configurations when Li_2S_2 -mixed discharge products exist are shown by the following equations:

Initially, during discharge, S_8 undergoes conversion to solid-phase polysulfide intermediates [Equation 1]. The formed polysulfide intermediates gradually undergo reduction to form Li_2S_2 [Equation 2]. The Li_2S_2 species can follow two possible routes: (a) A portion of Li_2S_2 disproportionates into Li_2S and S_8 [Equation 3] OR (b) Another portion of Li_2S_2 is reduced to Li_2S but not completely due to sluggish reaction kinetics [Equation 4]. We speculate that these reactions occur in a non-stepwise manner due to sluggish conversion kinetics in the solid-state. As a result, the discharge curve of an all-solid-state Li-S battery typically shows a single plateau, despite the existence of different phases. However, when the reaction kinetics is improved by increasing the temperature, two discharge plateaus appear in the discharge curve, indicating the formation of distinct polysulfide species, as shown in **figure 4a** below. Overall, the proposed electrochemical redox process from Equation 1 to 4 provides a possible explanation for the observed behavior in all-solid-state Li-S batteries. However, further investigation is necessary to validate and fully understand the underlying mechanisms of these reactions.

Fig. 4: (a) Voltage profile of ASSLSB tested at 60°C, with enlarged discharge profile shown between 1.9-2.1 V (vs. Li^+/Li).

8. *The addition of Lil to promote the oxidation process in Li-S batteries has been reported earlier. When the mixed discharge product with excess Li_2S_2 is formed, how does the Lil contribute to promoting the redox activities, than in the case of the formation of a single discharge product (Li_2S)? How the authors fixed the optimum quantity of Lil added to improve the oxidation? If they followed some previous attempts, the authors have to mention that.*

Response: We thank the reviewer for these important questions. To answer the first question, we conducted DFT calculations to investigate the contribution of Lil on promoting the redox activities of both Li_2S_2 and Li_2S , and our results are shown in **Figure 1d** and **Figure 1e** in the main manuscript. The lithium extraction energy was calculated to be +4.10 eV and +3.78 eV for Li_2S and Li_2S_2 molecules adsorbed on the Lil surface, respectively. Furthermore, the overall activation peak for Li_2S_2 was calculated to be 0.42 eV/atom lower as compared to Li_2S . While Li_2S_2 has better redox activity than Li_2S (according to our DFT calculations), this does not mean that a discharge product with an Li_2S_2 -dominant phase will be fully reversible. In other words, the delithiation process of Li_2S_2 is still hindered, which is why we look to facilitate this reaction through the addition of Lil. It is also important to note that sluggish conversion kinetics precludes the possibility of a single discharge product (Li_2S). While the dominant discharge product phase can be controlled by manipulating the lower potential limit, a single discharge product, whether that is Li_2S_2 or Li_2S , is unlikely to form experimentally because of the sluggish conversion from Li_2S_2 to Li_2S and Li_2S_2 disproportionation. In our study, we induce the formation of a Li_2S_2 -dominant discharge product but acknowledge that some Li_2S will also be formed. Therefore, both Li_2S_2 and Li_2S will contribute to irreversibility capacity in all-solid-state Li-S batteries.

To provide further clarification, we have completely re-written paragraph 3 of the results section to provide a detailed investigation regarding Lil and its contribution to promoting the redox activities of Li_2S_2 and Li_2S .

REVISION (Page 7): "Another recurrent feature observed in the literature regarding ASSLSBs is their poor electrochemical reversibility, particularly following the initial

discharge cycle.^{15,20,41} This phenomenon has been attributed to the irreversible formation of Li_2S , where the stable antifluorite structure of Li_2S necessitates high activation potentials, typically approaching 4V (versus Li^+/Li), to facilitate the electrochemical oxidation (or delithiation) of Li_2S back to S_8 during the charging process.^{42–45} We conducted density functional theory (DFT) calculations to investigate the influence of Li_2S_2 and Li_2S on the reversibility of ASSLSBs (**Supplementary Note 1**). The calculated formation energies of Li_2S_2 and Li_2S were approximately -1.01 eV/atom and -1.59 eV/atom, respectively (**Fig. 1d**). These results indicate that Li_2S_2 exhibits better redox activities compared to Li_2S . The formation energy of Li_2S_2 , however, remains considerably lower than that of S_8 . This suggests that both Li_2S_2 and Li_2S hinder the electrochemical reversibility of ASSLSBs. Previous studies have demonstrated the effective use of lithium iodide (LiI) to enhance the electrochemical oxidation of Li_2S , thereby achieving fully reversible ASSLSBs.^{16,46,47} Indeed, our DFT calculations reveal that the molecular conversion of $\text{Li}_2\text{S}_2/\text{Li}_2\text{S}$ to S_8 on the LiI(100) surface requires a lower activation barrier compared to the process in vacuum (**Fig. 1d, e**). These results suggest that LiI can facilitate the electrochemical oxidation of not only Li_2S but also Li_2S_2 , thereby improving the reversibility of ASSLSBs as a result. Further discussion regarding the DFT calculations and the role of LiI in promoting the electrochemical oxidation of $\text{Li}_2\text{S}_2/\text{Li}_2\text{S}$ is provided in **Supplementary Note 1.**"

Supplementary Note 1: The formation energies of Li_2S_2 , Li_2S and S were calculated in vacuum and on the LiI surface to provide atomic level insights regarding the catalytic mechanism of LiI. An energy correction of -0.503 eV/atom for anionic sulfur in the structure was utilized based on a previous study conducted by Persson et al.¹⁰ The calculated formation energy values for Li_2S_2 , Li_2S and S in vacuum are in close agreement with the data available in the Materials Project database.¹¹ A slight difference was observed due to the use of GGA in this work, as compared to a mixture of GGA and GGA+U in the Materials Project Database for modelling exchange-correlation.¹² The calculated values for formation energies in vacuum are also in good agreement with a previous study conducted by Wang et al.¹³ In vacuum, Li_2S has a formation energy of approximately -1.59 eV/atom and is shown to exist in the bulk phase rather than the molecular phase. This indicates that Li_2S is very stable and

requires a large activation potential to facilitate its electrochemical oxidation back to S during charge. The calculated formation energy of Li_2S_2 and Li_2S on the $\text{LiI}(100)$ surface suggests that the compounds are thermodynamically stable in their molecular form on $\text{LiI}(100)$, as compared to the compounds in vacuum. In contrast, the formation energy of S_8 on the $\text{LiI}(100)$ surface is calculated to be approximately +0.02 eV/atom. This indicates that the S_8 molecule breaks off to form a bulk sulfur phase after it has been oxidized from Li_2S to S on the $\text{LiI}(100)$ surface. These results demonstrate that LiI can facilitate the electrochemical oxidation of Li_2S_2 and Li_2S during charge.

To understand the delithiation process of Li_2S_2 and Li_2S during charging, the energy required to extract one lithium atom from bulk Li_2S and adsorbed Li_2S_2 and Li_2S was computed using DFT. Barrier to Li removal could not be calculated for bulk Li_2S_2 because it is meta-stable (energy above convex hull = 0.315 eV/atom), resulting in immediate structural decomposition.¹⁴ For the adsorbed substrate, one Li atom was removed from the adsorbed Li_2S and Li_2S_2 molecule as per the following reaction: (1) $\text{Li}_2\text{S} \rightarrow \text{Li-S}^* + \text{Li}^*$ and (2) $\text{Li}_2\text{S}_2 \rightarrow \text{Li-S-S}^* + \text{Li}^*$, respectively. The lithium extraction energy for the adsorbed Li_2S_2 and Li_2S molecules were calculated using the following expression: $E_{\text{ext}} = E(\text{Li-S}^* / \text{Li-S-S}^*) - E(\text{Li}_2\text{S}(\text{ads}) / \text{Li}_2\text{S}_2(\text{ads}))$. The lithium vacancy formation energy of a bulk Li_2S crystal was calculated using a large supercell of 96 atoms for bulk Li_2S , using the following expression: $E_{\text{vac}} = E(\text{Li}_2\text{S}(\text{bulk with 1 Li vacancy})) - E(\text{Li}_2\text{S}(\text{bulk}))$. The lithium extraction energy was calculated to be +4.10 eV and +3.78 eV for the adsorbed Li_2S and Li_2S_2 molecule, respectively, while the lithium vacancy formation energy of the bulk Li_2S crystal was calculated to be +5.75 eV per Li atom. After accounting for the energy required to form Li_2S on the LiI surface, the overall activation peak was lowered by 0.65 eV/atom, which indicates that the energy required for delithiation of Li_2S becomes lower in the presence of LiI as compared to bulk Li_2S . On $\text{LiI}(100)$ surface, the overall activation peak for Li_2S_2 as compared to Li_2S was 0.42 eV/atom lower.”

To answer the second question, previous studies have successfully employed LiI to enhance reaction kinetics in ASSLSBs and improve their reversibility/performance (*ACS Appl. Energy Mater.* 2022, 5, 8, 9429–9436; *Small* 2023, 2302179; *Adv. Funct. Mater.*

2022, 32, 2106174). However, these studies commonly employ Li_2S as the initiating active material and utilize LiI to form $\text{Li}_2\text{S-LiI}$ solid solutions. Furthermore, typically over 20 wt% of LiI is used to form these solid solutions. In our study, which employs S_8 rather than Li_2S as the starting active material, we sought to utilize as little of LiI as possible to maximize the active material content in the cathode composite.

To fix the optimum quantity of LiI to improve oxidation, we tested different ratios and determined that a minimum of 6 wt% LiI was necessary to attain fully reversible ASSLSBs. Lowering the LiI content to 3 wt% resulted in ASSLSBs that could only be charged to approximately 86%. The voltage profiles of ASSLSBs assembled with 3 wt% and 6 wt% LiI are shown in **Figure R4** for comparison.

Figure R4. (a) Voltage profile showing the reversibility of ASSLSBs in the first cycle with 3 wt% LiI . (b) Voltage profile showing the reversibility of ASSLSBs in the first cycle with 6 wt% LiI .

To provide further clarification on how we fixed the optimum quantity of LiI to improve oxidation, we have included **Figure R4** as **supplementary figure 12** and expanded the discussion in the revised manuscript.

REVISION (Page 13): "As for fixing the optimum quantity of LiI to facilitate the electrochemical oxidation of $\text{Li}_2\text{S}_2/\text{Li}_2\text{S}$, we determined that a minimum of 6 wt% LiI was necessary to attain fully reversible ASSLSBs when setting the lower voltage limit to 1.2V (vs. Li^+/Li). Lowering the LiI content to 3 wt% resulted in ASSLSBs that could only be charged to approximately 86% of discharge capacity (**Supplementary Figure 13**)."

9. In the rate performance of the Li-S configuration with excess Li₂S₂ component, the initial capacity of the sample at 0.2 A g⁻¹ is 933 mAh g⁻¹, which is improved at higher current densities. The authors provided the explanation for the gradual improvement in improvement in the capacity of the sample with the aid of two processes (1) Initial sulfur activation; 2) The electrochemically active discharge product of LGPS solid electrolyte). However, after reaching the current rate of 6.0 A g⁻¹ (303.7 mAh g⁻¹), the recovering capacity at 0.2 A g⁻¹ has been reported as 1222.4 mAh g⁻¹. Which is much higher than all other obtained capacities. The aforementioned two explanations are not sufficient to explain this high-recovered capacity obtained in the sample since the sulfur activation and SE dissociation are the initial processes. The authors have to give some explanations for this.

Response: We thank the reviewer for this question. The cycling results shown in **Figure 3d** of the main manuscript show that the capacity continues to gradually increase even after 200 cycles. Furthermore, **Supplementary Figure 16** shows that the polarization decreases from 0.721V in the first cycle to 0.682V in the 200th cycle. These results suggest that the sulfur activation and SSE dissociation process continues to occur beyond 45 cycles (i.e., cycle number of the rate performance tests). Although the recovering capacity at 0.2 A g⁻¹ is indeed much higher than all other obtained capacities, this result is consistent with our cycling data. A similar phenomenon can be observed in a recent report studying all-solid-state Li-S batteries (Small 2023, 2300420).

Supplementary Figure 16. Voltage profile of ASSLSB at 1st, 100th, and 200th cycle.

10. The overall style of presentation and the many sentences in the manuscript has to be further modified.

Response: We agree with the reviewer's assessment that there is ample opportunity to enhance the overall style of presentation in this manuscript. In our revised version, we have diligently addressed your concerns and incorporated your invaluable suggestions and comments to elevate the quality of the paper. We sincerely appreciate the reviewer's constructive feedback and invaluable insights, which have greatly contributed to the improvement of this work.

The authors have to address the aforementioned points before the consideration of the manuscript for publication.

Reviewer #2 (Remarks to the Author):

All-solid-state lithium-sulfur (Li-S) batteries are regarded as the most promising energy storage system because of their low cost and theoretically high energy density. However, the mechanism of all-solid-state Li-S batteries remains elusive. The study by Kim et al. investigates the root cause of the lower practical discharge capacities of all-solid-state Li-S batteries than their theoretical value. By virtue of advanced characterization techniques such as XAS and ToF-SIMS, the authors identified that the mixed discharge product of $\text{Li}_2\text{S}_2/\text{Li}_2\text{S}$ accounts for the lower practical discharge capacities. DFT calculations further confirms that Li_2S_2 has better electrochemical kinetics than Li_2S . Based on this understanding, the authors devise a strategy that involves modifying the lower cut-off potential to promote a Li_2S_2 -dominant discharge product and incorporating LiI into the cathode composite. This approach results in fully reversible ASSLSBs with outstanding cycling stability for over 1500 cycles at room temperature. In summary, this work represents a significant advancement in ASSLSB research by offering novel insights into the discharge products of ASSLSBs and demonstrating unprecedented cycling results. The strategy presented in this study is highly practical. Therefore, I would like to recommend it for publication in the prominent journal Nature Communications. Also, there are some minor comments, as follows:

Response: We sincerely thank the reviewer for their positive rating and recommendation to publish our work in **Nature Communications**. Below, we provide a detailed response addressing each of your valuable and constructive suggestions.

1. *The authors propose that by manipulating the lower cut-off voltage and inducing a discharge product dominated by Li_2S_2 , the reversibility and cycling stability of ASSLSBs can be enhanced. I suggest authors to provide more data with different lower cut-off voltages to strengthen their claim.*

Response: We thank the reviewer for this constructive suggestion. Indeed, it is important to demonstrate experimentally the influence of a Li_2S_2 -dominant discharge product on the cycling stability of ASSLSBs. We assembled two ASSLSBs and tested them using a lower cut-off voltage of 1.2V and 0.4V (vs. Li^+/Li) for comparison. The voltage profile and cycling data of the ASSLSBs are shown in Figure R1 below.

Figure R3. (a) Voltage profile of ASSLSBs tested using different lower limit potentials. (b) Corresponding cycling performance of ASSLSBs tested at 0.2 A g⁻¹ and 25°C.

Lowering the battery's potential to 0.4 V enhances its initial discharge capacity to approximately 1300 mAh g⁻¹, compared to around 850 mAh g⁻¹ for the battery discharged to 1.2 V (**Figure R3a**). The higher initial discharge capacity of the 0.4 V cell can be attributed to an increased formation of Li₂S. Furthermore, both cells contain 6 wt% LiI, however, only the 1.2 V discharged cell is fully reversible on charge, while the 0.4 V discharged cell only reaches about 80% of charge capacity. This suggests that more than 6 wt% LiI is needed to electrochemically address the extra Li₂S that is formed.

A mixed discharge product with dominant Li₂S₂ phase enhances long-term cycling stability due to two main reasons. First, Li₂S₂ exhibits better kinetics compared to Li₂S, as supported by our DFT results. This improved kinetics leads to enhanced reversibility, contributing to better cycling performance. Second, the volume expansion of Li₂S₂ is lower than that of Li₂S, with Li₂S₂ expanding by approximately 60% compared to Li₂S's expansion of around 80%. This reduced volume expansion helps alleviate the volume-induced failures that can occur during cycling. **Figure R3b** illustrates the significant improvement in cycling stability observed in the cell with a Li₂S₂-dominant discharge product.

To provide further clarification on the advantages of a Li₂S₂-dominant discharge product, we have included **Figure R3** as **supplementary figure 12** and expanded the discussion in the revised manuscript.

REVISION (Page 13): “Although promoting a discharge product with an Li_2S_2 -dominant phase comes at the expense of the initial discharge capacity, doing so enhances both the reversibility and cycling stability of ASSLSBs (**Supplementary Fig. 12**). This is because Li_2S_2 is more electrochemically active than Li_2S , and the volumetric expansion of Li_2S_2 is comparatively smaller, at ~60%, compared to Li_2S which expands by approximately 78%.”

2. Why did the authors specifically choose 6 wt% Lil? The effect of Lil content should be investigated.

Response: We thank the reviewer for this important question. Previous studies have successfully employed Lil to enhance reaction kinetics in ASSLSBs and improve their reversibility. These studies commonly utilize Lil to form solid solutions, with a typical Lil content exceeding 20 wt% in the cathode composite. Our study aimed to identify a feasible approach for enhancing the electrochemical performance of ASSLSBs. Consequently, we sought to utilize as little of Lil as possible to achieve this improvement. We determined that a minimum of 6 wt% Lil was necessary to attain fully reversible ASSLSBs. For example, lowering the Lil content to 3 wt% resulted in ASSLSBs that could only be charged to approximately 86%. The voltage profiles of ASSLSBs assembled with 3 wt% and 6 wt% Lil are shown in **Figure R4** for comparison.

Figure R4. (a) Voltage profile showing the reversibility of ASSLSBs in the first cycle with 3 wt% Lil. (b) Voltage profile showing the reversibility of ASSLSBs in the first cycle with 6 wt% Lil.

To provide further clarification on how we fixed the optimum quantity of Lil to improve oxidation, we have included **Figure R4** as **supplementary figure 12** and expanded the discussion in the revised manuscript.

REVISION (Page 13): "As for fixing the optimum quantity of Lil to facilitate the electrochemical oxidation of $\text{Li}_2\text{S}_2/\text{Li}_2\text{S}$, we determined that a minimum of 6 wt% Lil was necessary to attain fully reversible ASSLSBs when setting the lower voltage limit to 1.2V (vs. Li^+/Li). Lowering the Lil content to 3 wt% resulted in ASSLSBs that could only be charged to approximately 86% of discharge capacity (**Supplementary Figure 13**)."

3. Does Lil contribute any capacity during the charge/discharge process?

Response: We thank the reviewer for raising this important question. To determine how much capacity Lil contributes during the charge/discharge process, we assembled cells using cathode composites using Lil as the active material, as shown in **Figure R5** below. The results show that the capacity contribution stemming from Lil amounts to less than 10 mAh g^{-1} . Therefore, we conclude that Lil contributes negligible capacity during the charge/discharge process, especially considering the relatively small quantity of Lil (6 wt%) incorporated into the cathode composite for the full cells.

Figure R5. Voltage profile showing the reversibility capacity of Lil.

4. What are the reasons for the large electrochemical polarization in discharge curves at low temperature in Figure 4e?

Response: We thank the reviewer for this question. Both liquid and solid-state Li-S batteries exhibit significant electrochemical polarization due to the low electronic/ionic conductivity of sulfur and its discharge products ($\text{Li}_2\text{S}_2/\text{Li}_2\text{S}$). When ASSLSBs are tested at -10 degrees, charge transfer resistance increases, leading to reduced capacity and higher overpotential, as depicted in **Figure 4e** of the main manuscript. Moreover, our study employed a Li/In alloy anode, and at -10 degrees, the alloying reaction between these two materials is likely impeded, resulting in the distinct discharge curve that is observed in the initial cycle.

Figure 4: (e) Voltage profile of ASSLSB tested at 0.2 A g^{-1} and -10°C .

5. Why are there two plateaus in the discharge curves when testing 60°C ?

Response: We appreciate the reviewer's question. The appearance of an additional distinct plateau in the discharge curve when the cell is tested at 60 degrees can be attributed to enhanced charge transfer kinetics at higher temperatures, enabling a smoother progression of sulfur redox reactions. This phenomenon suggests the formation of intermediate sulfur species, which likely accounts for the observed plateau. We discuss this in more detail on page 16 of our manuscript.

(Page 16): “When tested at 60°C , the cell shows much lower overpotential, and delivers a high initial discharge capacity of $1136.8 \text{ mAh g}^{-1}$ (**Fig. 4a**). Interestingly, another discharge plateau appears at approximately 2.0 V (vs. Li^+/Li). At 25°C , the stepwise

transition from elemental sulfur to high order polysulfides, low order polysulfides, and finally Li_2S is not obvious in a solid-state configuration, as a high conversion barrier results in sluggish conversion kinetics. However, charge transfer within the cathode composite is improved under more favorable conditions such as at elevated temperature, allowing for stepwise sulfur redox to occur. This likely gives rise to a distinct discharge plateau in the voltage profile at 60°C , corresponding to the formation of intermediate sulfur species. A similar phenomenon has been observed in ASSLSBs that incorporate selenium into the sulfur cathode and solid-state lithium-selenium batteries, which is logical considering the high conductivity of selenium.^{64,67}

6. *Typo in figure 2 caption 'cia'*

Response: We thank the reviewer for bringing this to our attention. We have fixed this error in the manuscript.

7. *There are two figure 4's in this manuscript – the latter should be figure 5*

Response: We thank the reviewer for bringing this to our attention. We have fixed this error in the manuscript.

REVIEWERS' COMMENTS

Reviewer #1 (Remarks to the Author):

The authors addressed and executed most of the comments in the revised manuscript. The article may be accepted in its present form.

Reviewer #2 (Remarks to the Author):

The authors have effectively addressed all the concerns raised by the reviewers. These changes have significantly strengthened the paper and have enhanced its overall clarity and coherence. Therefore, now I would like to recommend it for publication in Nature Communications.